# Machine Learning Detects Pattern of Differences in Functional Magnetic Resonance Imaging (fMRI) Data between Chronic Fatigue Syndrome (CFS) and Gulf War Illness (GWI)

**DOI:** 10.3390/brainsci10070456

**Published:** 2020-07-17

**Authors:** Destie Provenzano, Stuart D. Washington, Yuan J. Rao, Murray Loew, James Baraniuk

**Affiliations:** 1School of Medicine, Georgetown University, Washington, DC 20057, USA; djp92@georgetown.edu (D.P.); sdw4@georgetown.edu (S.D.W.); 2School of Engineering and Applied Science, George Washington University, Washington, DC 20052, USA; loew@email.gwu.edu; 3School of Medicine and Health Sciences, George Washington University, Washington, DC 20052, USA; yrao@mfa.gwu.edu

**Keywords:** Gulf War illness (GWI), chronic fatigue syndrome (CFS), machine learning, ensemble model, fMRI, BOLD, random forest, logistic regression, decision tree, adaboost, QDA, SVM, naïve Bayes, neural net

## Abstract

Background: Gulf War Illness (GWI) and Chronic Fatigue Syndrome (CFS) are two debilitating disorders that share similar symptoms of chronic pain, fatigue, and exertional exhaustion after exercise. Many physicians continue to believe that both are psychosomatic disorders and to date no underlying etiology has been discovered. As such, uncovering objective biomarkers is important to lend credibility to criteria for diagnosis and to help differentiate the two disorders. Methods: We assessed cognitive differences in 80 subjects with GWI and 38 with CFS by comparing corresponding fMRI scans during 2-back working memory tasks before and after exercise to model brain activation during normal activity and after exertional exhaustion, respectively. Voxels were grouped by the count of total activity into the Automated Anatomical Labeling (AAL) atlas and used in an “ensemble” series of machine learning algorithms to assess if a multi-regional pattern of differences in the fMRI scans could be detected. Results: A K-Nearest Neighbor (70%/81%), Linear Support Vector Machine (SVM) (70%/77%), Decision Tree (82%/82%), Random Forest (77%/78%), AdaBoost (69%/81%), Naïve Bayes (74%/78%), Quadratic Discriminant Analysis (QDA) (73%/75%), Logistic Regression model (82%/82%), and Neural Net (76%/77%) were able to differentiate CFS from GWI before and after exercise with an average of 75% accuracy in predictions across all models before exercise and 79% after exercise. An iterative feature selection and removal process based on Recursive Feature Elimination (RFE) and Random Forest importance selected 30 regions before exercise and 33 regions after exercise that differentiated CFS from GWI across all models, and produced the ultimate best accuracies of 82% before exercise and 82% after exercise by Logistic Regression or Decision Tree by a single model, and 100% before and after exercise when selected by any six or more models. Differential activation on both days included the right anterior insula, left putamen, and bilateral orbital frontal, ventrolateral prefrontal cortex, superior, inferior, and precuneus (medial) parietal, and lateral temporal regions. Day 2 had the cerebellum, left supplementary motor area and bilateral pre- and post-central gyri. Changes between days included the right Rolandic operculum switching to the left on Day 2, and the bilateral midcingulum switching to the left anterior cingulum. Conclusion: We concluded that CFS and GWI are significantly differentiable using a pattern of fMRI activity based on an ensemble machine learning model.

## 1. Background

Gulf War Illness (GWI) and Chronic Fatigue Syndrome (CFS) are two debilitating disorders characterized by chronic widespread pain, fatigue, sleep abnormalities, and cognitive impairment that are worsened by mild to moderate exertion (post-exertional malaise or exertional exhaustion) [1,2,3,4,5].

GWI has been linked to military exposures to nerve agents, pyridostigmine bromide pills, and other potential neurotoxins and affects 25–30% of the 700,000 individuals who served in the 1990–1991 Persian Gulf War [6,7,8]. CFS is estimated to affect at least 1 million individuals in the United States and causes USD 9.1 billion in annual losses in productivity [2]. Despite sharing similar symptoms, the pathological mechanisms for both remain unknown and controversy persists as to the underlying physiology [4,8,9,10].

Functional Magnetic Resonance Imaging (fMRI) has emerged as a promising methodology to differentiate both GWI and CFS from corresponding sedentary controls (SCs) by examinations of regions that are activated or deactivated at rest or during tasks, and changes in brain activation that are caused by exertion [11,12,13,14,15,16,17,18,19,20,21,22]. Our group previously established that differences persisted between GWI and a sedentary control (SC) on an fMRI scan after a mild exercise stress test [11,23]. Using a similar experimental setup, our group was also able to differentiate Chronic Fatigue Syndrome (CFS) [24] and GWI [25] from a sedentary control before and after exercise using machine learning (a Logistic Regression algorithm). We also found that CFS was differentiable from GWI after exercise using fMRI data collected in a similar fashion [26]. We hypothesized that, by using a similar experimental set up, fMRI data from both GWI and CFS and machine learning could be used to differentiate GWI from CFS as well.

We modeled the fMRI differences previously detected through exertional exhaustion by conducting the continuous version of the n-back working memory test before (Day 1) and after (Day 2) a bicycle exercise stress test. The Blood Oxygenation Level Dependent (BOLD) signals were compared between GWI and CFS subjects on both days. Our approach utilized a standard predictive model build involving feature extraction, feature selection, model selection, validation, and evaluation of performance [27]. SPM12 was used to map activations from each subject to the Automatic Anatomical Labeling (AAL) atlas.

Separate models were created for pre- (Day 1) and post- (Day 2) exercise periods by splitting data into a training and testing set. We passed features through a K-Nearest Neighbor algorithm (KNN), Linear Support Vector Machine (SVM), Decision Tree, Random Forest, AdaBoost, Naïve Bayes Classifier, Quadratic Discriminant Analysis (QDA), Logistic Regression, and Neural Net (Multilayer Perceptron) to create an ensemble model that generated a higher predictive power across the regions selected by all of the above. The model results were combined, analyzed for statistical measures such as sensitivity and the False Discovery Rate (FDR), and subjected to a shuffle test to ensure performance. Multiple metrics were used to further account for sample imbalance.

The outcome of this model was a multivariate pattern of AAL regions activated in BOLD fMRI data that differentiate CFS from GWI. This difference indicates that, despite similar symptoms, they have a pattern of different BOLD activation patterns that suggests they are separate diseases. This study is the first of its kind to attempt to differentiate CFS from GWI using machine learning and confirms that CFS and GWI are differentiable using a pattern of fMRI BOLD data and the n-back working memory task.

## 2. Methods

### 2.1. Ethics

Subjects provided written informed consent for participation and the use of all data for publication purposes. Studies were approved by the Georgetown University Institutional Review Board (IRB 2009-229, 2013-0943, 2015-0579) and the U.S. Army Medical Research and Material Command (USAMRC) Human Research Protection Office (HRPO A-155547.0, A-18749), and registered on clinicaltrials.gov as NCT01291758, NCT03560830, and NCT03567811. All clinical investigations were conducted according to the principles expressed in the Declaration of Helsinki.

### 2.2. Approach

Patient fMRI data were collected from subjects who fit the criteria for CFS or GWI before and after exercise and were run through a series of machine learning models to attempt to differentiate the two. An ensemble modeling approach, using multiple predictive models, was selected because of the potential increased accuracy and lower likelihood of overfitting.

Ensemble learning in machine learning is the process of using multiple predictive algorithms to obtain a better predictive performance than could be obtained from any one learning model alone [28,29]. Ensemble modeling techniques assist in minimizing noise, bias, and variance in data by providing a multifaceted approach to feature selection [30]. This study chose to adopt an ensemble model based on the output of features selected by a Nearest Neighbor algorithm, Linear SVM, Decision Tree, Random Forest, AdaBoost, Naïve Bayes Classifier, QDA, Logistic Regression, and Neural Net.

To determine the features to use in the ensemble model, this study used a standard predictive model build process of feature extraction, feature selection, model selection, validation, and evaluation of performance. (Figure 1) A standardized brain atlas able to represent both sides of the brain was selected to define the parameters for each feature so that the bounds of the model could be easily replicated in future studies and used to predict CFS vs. GWI. The Automatic Anatomical Labeling (AAL) atlas was selected due to its widespread use, easy generalizability and accessibility across multiple types of software and platforms, and good representation of the physical and functional areas of the brain. The AAL atlas additionally contains cerebellar and cortical regions, which the classic Brodmann cortical and SUIT cerebellar atlases lack [31]. Simplistic ensemble modeling techniques involve evaluating the average, mode, or weighted average of the results of multiple models, whereas advanced techniques may seek to evaluate bagging (aggregate multiple weak models to obtain a better prediction) or boosting (sequential training of weak learners to transition them to strong learners). This study chose to combine methods similar to both “stacking” and the “Bucket of Models” results to find the best ensemble option. Stacking seeks to create a combined model output where multiple models may be input into this final algorithm [32]. Bucket of Models seeks to train data on multiple models and then select the best performing model based on end results [33].

To find the best optimal final Bucket of Models, we sought to report the combination of AAL regions that resulted in a threshold of 69% or greater final accuracy in predictive power of GWI vs. CFS across all of the “Buckets” of models. This threshold of 69% was selected to attempt to include the results of more models and potential selected AAL regions for later filtering using the combined modeling technique.

Predictions generated this way are then typically combined through “stacking”, where one final combining algorithm is used to generate predictions [34]. Instead of using one final predictive “combiner” algorithm, our approach modified this procedure to use all of the algorithms in tandem and investigate the accuracy within the regions of overlap. Here, a region of overlap or segment of subjects was defined to be a prediction selected by one or more models.

Data from both fMRI collection days (before and after exercise) were used to create two separate models because earlier results had found that CFS and GWI were differentiable respectively before and after exercise when compared to a sedentary control. For this study, we aimed to test if these same methods could be used to differentiate CFS and GWI when compared to each other.

A sedentary control, who did not meet criteria for either disease, was not used for this analysis, as the focus of this study was to identify potential differences between CFS and GWI using machine learning. We previously published results showing the differences found using machine learning between CFS and a sedentary control [24] and GWI and a sedentary control [25]. A table is available in the Appendix A of this paper showing how the regions selected by this model for CFS against GWI for each day compares to the regions selected by these prior models against the sedentary controls on each day (Appendix A). A table detailing uncommon abbreviations and terms used in this paper is available in the Appendix A (Appendix A). 

### 2.3. Subjects

Candidates for the study responded online, by phone, or personal contact. Two hundred and sixteen subjects underwent telephone screening after verbal consent. Of these, 105 declined to participate or were excluded from participation after protocol explanation and the assessment of chronic medical and psychiatric disease [35,36]. Chronic Fatigue Syndrome was assessed by the 1994 Fukuda CDC criteria as having 6 months of debilitating fatigue without medical or psychiatric cause plus at least four of the following eight criteria: problems with memory or concentration, sore throat, sore lymph nodes, myalgia, arthralgia, headache, sleep disturbance, and post-exertional malaise [3]. GWI was confirmed by examination using the Chronic Multisymptom Illness (CMI) and Kansas criteria [1,3]. The Kansas criteria require moderate or severe chronic symptoms in at least three of six domains: fatigue/sleep, muscle/joint pain, neurological/cognitive/mood, gastrointestinal, respiratory, and skin symptoms [1].

Subjects were admitted to the Georgetown Howard Universities Clinical Translation Science Clinical Research Unit. Subjects were tested with the n-back working memory task in a 3T MRI scanner on two separate days. Subjects had their first fMRI scan during the n-back working memory task after an overnight rest, and the second after a submaximal exercise stress test. Subjects cycled at 70% of their age-predicted maximum heart rate (220 – age) for 25 min, then ramped up their effort to reach 85% of their predicted heart rate.

This study reports on 38 subjects with Chronic Fatigue Syndrome and 80 with Gulf War Illness. Subject demographics are reported here. Full subject data and performance results from the n-back working memory task between the two groups are available and published elsewhere and in the online Appendix A [11,37,38,39,40].

### 2.4. Magnetic Resonance Imaging (MRI) Data Acquisition

A Siemens 3T Tim Trio Scanner equipped with a transmit–receive body coil and a commercial 12-channel head coil array was used to perform neuroimaging. The image parameters for the structural 3D T1-weighted Magnetization Prepared Rapid Acquisition Gradient Echo (MPRAGE) image parameters were: Repetition Time/Echo Time (TR/TE) = 1900/2.52 ms, T1 = 900 ms, Field-of-View (FoV) = 250 mm, 176 slices, slice resolution = 1.0 mm, and a voxel size of 1 × 1 × 1 mm. Functional T2*-weighted gradient Echo Planar Imaging (EPI) parameters were: number of slices = 47, TR/TE = 2000/30 ms, flip angle = 90°, matrix size = 64 × 64, FoV = 205 mm^2^, and voxel size = 3.2 mm^2^ (isotropic).

### 2.5. Data Pre-Processing

The CONN version 17 toolbox was used to preprocess BOLD data through a built-in software default pipeline [41]. To preprocess the data, a series of steps was taken that included: (a) spatial smoothing with a 6 mm Full Width at Half Maximum (FWHM) Gaussian filter, (b) slice-timing correction, (c) the realignment and unwarping of the functional images, (d) the spatial normalization of the anatomic scan to the Montreal Neurological Institute (MNI) standard stereotactic space of a 2.0 mm^3^ (isotropic) voxel size, and (e) artifact detection tool automatic outlier detection for framewise displacement [42]. Preprocessed EPI data were then modeled for each subject using instruction, fixation, 0-back (as defined by the n-back task), and 2-back (as defined by the n-back task) [43]. We used a one-sample *t*-test to compare the 2-back > 0-back condition as our contrast of interest. We used motion parameters like translation and rotation as covariates of no interest. By comparing the high cognitive load 2-back to the low cognitive load “stimulus matching” 0-back task, this generated the 2-back > 0-back condition and was used to identify voxels that were significantly more activated.

### 2.6. Feature Extraction

Voxels were grouped into Automated Anatomical Labeling (AAL) regions as determined by Montreal Neurological Institute (MNI) coordinate mapping using a custom MATLAB program that employed functions from both SPM12 and XjView 9.6 [44]. The AAL atlas is a brain atlas commonly used in fMRI studies that divides three-dimensional regions of the brain into functional areas based on the MNI coordinate mappings [45]. The MNI coordinates are a template brain map that places scan data into the brain space based on x, y, and z locations [45]. By mapping the AAL atlas and MNI coordinates within SPM12, the authors were able to roughly define where voxels were in relation to neuroanatomy for every cohort in the same location without defining unique Regions of Interest (ROIs). Functional *t*-statistical maps were generated using the one-sample *t*-values from the 2-back > 0-back relative activation within the BOLD data of each subject. The total number of voxels activated was plotted as a function of *t*-values to determine the optimal threshold for comparison, as we had described in a previous study [25]. Ultimately, voxels with *t*-values exceeding 3.17 (*p* < 0.001 uncorrected) were selected because this maintained statistical significance while also allowing a large number of voxels per subject within each map. These remaining significant voxels were then grouped for each subject into the standard Automated Anatomical Labeling (AAL) atlas (Appendix A) [15]. The total number of voxels contained in each AAL region was considered a “feature” for input into the model. These features for each individual, or AAL regions, were the independent variables used in the model. This resulted in 117 features, due to the 117 AAL regions, for all subjects. The centers of mass, total voxels per region, and AAL identifier for each AAL region are included in the Appendix A. (Appendix A, Appendix A) [46].

A multistep feature reduction process was used to determine the total number of remaining significant features that were used for the final model estimation method. Pearson’s correlation coefficients were determined for each AAL region compared to every other AAL region to first assess if any regions were highly correlated with one another. The output was a matrix of AAL regions. This was done to assess multicollinearity, or regions of highly correlated inputs. Multicollinearity does not always change the overall predictive power of a model, but can render the calculations of individual predictors and any single variable invalid [47,48]. Instances of perfect multicollinearity (one predictor completely predicts another) cause the design matrix to have one less than the full rank and is then unable to be inverted, which makes the calculation of the ordinary least squares estimator impossible [49]. It is important to control for multicollinearity because it can create inaccurate algorithms, models that overfit the data, and large standard errors for individual coefficients [46,50]. Pearson’s correlation coefficients were calculated amongst all AAL regions three times on the training set alone, the testing set alone, and for the combined training and testing set to ensure all regions of perfect multicollinearity were removed prior to model building. For each dataset, this resulted in 117 × 117 initial correlations, with additional runs being done if any variables needed to be combined. Additional correlation matrices were generated as needed to ensure no newly generated features were recombinant features of any others. For this model, all features fed into the final model build had to be under the threshold of R < 0.9, otherwise they would be assumed to be linearly predictive of one another and were then combined, compared, and removed. The absolute value of Pearson’s correlation coefficient was used for this analysis, as the most important consideration is how strongly any variable can predict another variable.

Pearson’s correlation coefficients were additionally used to compare symptom severity to selected AAL regions. Symptom severity scores were compared by the Pearson’s correlation coefficient to the total voxels in each of the selected regions for the final predictive model for each group. The matrix of correlation coefficients showcasing the difference between Pearson’s correlation coefficients for GWI and CFS for each symptom vs. selected AAL regions was then plotted for each day.

### 2.7. Feature Selection and Predictive Model Building

Each AAL region was subjected to multiple rounds of feature selection, elimination, and then the ultimate determination of importance and input into the model. The dataset was first partitioned into separate training and testing sets to build the predictive models. The testing set here functions as a validation, or hold-out sample, that is different and does not overlap with the training sample used to build the model. Thus, the training data used to build the model were kept separate from the validation/testing data used to report the results.

Prior to model building to account for potential sample imbalance, the dataset was evaluated at several partitions at respective 70:30, 60:40, and 50:50 splits into training and testing data. This was done because of the small sample size and greater number of GWI subjects compared to CFS subjects. Sample imbalance in extreme cases can prevent the model from learning and can create highly inaccurate estimates that predict that all outcomes are one class [30]. The number of samples that remained at each split was examined by looking at the total subjects in each split and by a preliminary set of test models. It was decided to use the 50:50 split, as this provided ample data to validate, allowed for more CFS and GWI individuals to be present in both the training and testing sets, and produced statistically significant results. These multiple “test” training and testing splits were not used in the final model build, as this was a preliminary step, meaning that no data from the testing set were used to train the final set of models. The final dataset used for this manuscript was a 50:50 training to testing set split, where the training sample was used to build the model on 50% of the data, and the testing set was used to validate the model on the remaining 50%.

Special concern in selecting this split was given to the small sample size, as a small validation set could bias the models towards overfitting.

AAL regions not excluded by the Pearson’s correlation coefficient analysis were passed through the built-in scikit learn Recursive Feature Elimination (RFE) algorithm [51]. Recursive Feature Elimination (RFE) is a greedy feature selection algorithm that is similar to sequential backward selection in a stepwise Logistic Regression. A greedy algorithm, like RFE or a Logistic Regression, adds and removes new features until it determines the best local optimized model at that point in time.

Multiple attempts were made to scale, increase, and reduce the feature list to best identify input variables. For example, to further identify regions of importance, we binned each variable into quartiles (Q1: < 25%, Q2: 25–50%, Q3: 50–75%, Q4: > 75%). These binned variables were included in model creation to attempt to identify non-parametric effects in the outcome variable. Although these binned variables did not remain in the final model, they are mentioned here as an example of the techniques tested.

The remaining feature list was then passed through the built-in scikit learn K-Nearest Neighbors algorithm, Linear SVM algorithm, Decision Tree, Random Forest, AdaBoost algorithm, Naïve Bayes Classifier, Quadratic Discriminant Analysis algorithm, Logistic Regression, and Neural Net. Each of the models in the “Bucket of Models” was trained on the training set and validated on the testing set. So for a variable to remain in the final model, it had to be included in all of the algorithms and have importance in the Random Forest according to the reported importance metric.

The Random Forest Importance metric was automatically generated using the default scikit learn package. The metric indicates the strength that each variable has in the model for predicting the final outcome of the model. These importance metrics increase in value as the model increases in performance. For example, an important feature in a model that does not accurately predict the outcome is likely not predictive of the outcome. Conversely, an important variable in a highly predictive model is likely an accurate predictor of the outcome.

Features were first automatically and then manually reviewed for importance based on KNN inclusion, Linear SVM kernel feature importance (coefficients), Decision Tree inclusion, Random Forest Importance, AdaBoost inclusion, Naïve Bayes inclusion, QDA inclusion, Neural Net inclusion, and Logistic Regression coefficient. All features with an importance of 0 were removed from subsequent model builds. The model was iteratively re-built until all models were able to achieve an accuracy of 69% or greater based on the given inputs.

This approach, similar to a “Bucket of Models” approach, allowed for both high accuracy and generalizability, as models created from the features used in multiple models generally outperformed any single model working alone. Next, in an approach similar to “stacking,” predictions were compared across different models to determine the accuracy within regions of overlap, or segments of subjects. A separate analysis was undertaken to determine how accuracy and generalizability changed when a segment of subjects within the validation set was selected, or predicted to be an outcome state of CFS, by one or more models. For example, if a subject was selected, or predicted to be CFS, by any one model, it would be included in one or more category. Similarly, for the two or more, a subject had to be selected within the validation set by at least two models. The accuracy for each of these segments (one or more, two or more, three or more, etc.) was then compared to the total number of subjects selected in each of these segments. This was done because a highly accurate model that selected only 1% of the population would not be effective in practice. The accuracy in this analysis was determined by the total correct predictions of CFS vs. GWI status.

The final results and model outcome was a pattern of independent model features (AAL regions) that was able to predict CFS from GWI status.

The results were validated for statistical significance using a shuffle test, where labels on both the outcome variable and predictor variables were shuffled and then re-run through the model build multiple times to test if they could achieve a similar score at random. Although not always necessary in machine learning situations such as these, the Bonferonni correction was used to provide a conservative measure of correction for multiple correlations [52].

## 3. Results

### 3.1. Subjects

All subjects had a sedentary lifestyle, with less than 40 min of active aerobic work or exercise per week. The subjects spanned a similar age range, however, they had different distributions of gender and BMI. Thus, age, gender, and BMI were controlled for in the final model build. Both CFS and GWI had significantly worse symptoms [53] and quality of life [54] than age-matched sedentary controls (Table 1). Data showing the comparison of these GWI and CFS symptoms to a control are available in the Appendix A (Appendix A).

### 3.2. Feature Extraction

The Pearson’s correlation coefficient (R) was calculated for every subject across all AAL regions to define initial features and check for instances of multicollinearity. The Pearson’s correlation coefficients were < 0.9 (Figure 2) for all regions that would have overlapped. Therefore, all of the 117 regions were included in the feature elimination and selection step.

### 3.3. Feature Selection

Features were passed through an iterative process of model building and feature reduction based on inclusion and importance metrics. Thirty AAL regions on the first day and thirty-three AAL regions on the second day (Table 2, Figure 3, Figure 4 and Figure 5) were included as the final features for all models. Random Forest Importance metrics and corresponding SVM and Logistic coefficients are reported. All of the features included in the final set were included in each individual model.

Features were first automatically selected by the algorithms and then manually reviewed for coefficients and feature importance. Features that had a Random Forest Importance of “0” were removed on subsequent runs. For the final set of models, any features that had a standard error greater than the importance metric itself were also removed. The final list (Figure 4) is visually displayed in order below.

### 3.4. Model Results and Validation

A K-Nearest Neighbors algorithm (Day 1: 70%/Day 2: 81%), Linear SVM (Day 1: 70%/Day 2: 77%), Decision Tree (Day 1: 82%/Day 2: 82%), Random Forest (Day 1: 77%/Day 2: 78%), AdaBoost (Day 1: 69%/Day 2: 81%), Naïve Bayes (Day 1: 74%/Day 2: 78%), QDA (Day 1: 73%/Day 2: 75%), Logistic Regression model (Day 1: 82%/Day 2: 82%), and Neural Net (Day 1: 76%/Day 2: 77%) were able to differentiate CFS from GWI before and after exercise, respectively, with an average of 75% accuracy in predictions across all models before exercise and 79% after exercise. (Table 3, Appendix A)

Selections included by all nine models resulted in accuracies of 100% on Day 1 and 100% on Day 2 (Figure 5). However, selections were also included if any seven or more models also had an accuracy of 100% and captured a larger portion of the overall CFS population. In general, segments that were selected by fewer total algorithms and not included in the larger overlap group of models had a much lower accuracy. Although the average accuracy was 75% on Day 1 and 79% on Day 2 across all models, it can be inferred that if a region was selected by only one model it might have been more likely to be a false positive selection.

Labels were shuffled and randomized to the data and passed through the model building process 1000 times to simulate a shuffle test. No subsequent modeling attempt was able to achieve a similar score across all models for the individual model results on the randomized labels. As such, it was assumed the model performed at a *p* < 0.05 level. This indicated all models were statistically significant for the given variables.

Correction for multiple correlations was considered. When the Bonferroni correction was applied, the statistical significance remained, as no shuffled run was able to obtain similar results. The False Discovery Rate was also considered in light of these observations and was found to be optimized as well when any seven or more models selected a region.

Symptom severity scores for each individual were compared by Pearson’s correlation coefficients to total voxels in each of the selected regions for the final predictive model. The difference in these correlations was plotted for each AAL region vs. symptom severity score for GWI vs. CFS to further emphasize the relationship between physiological symptoms and the AAL regions implicated in the predictive model (Figure 6). The magnitude in differences for most of the symptoms was small, and regions implicated in the pain matrix (insula, parietal regions) and in areas involved in memory and attention showed the starkest contrast between GWI and CFS [23,24,25,26].

## 4. Discussion

This study demonstrated that two ensemble models applied to the two datasets before and after exercise were able to distinguish CFS from GWI status. We previously demonstrated that accuracy increased in a machine learning algorithm used to distinguish CFS [24] and GWI [25] from a sedentary control once the subjects had undergone a light exercise stress test. In this model, however, we obtained accuracies of 100% on both days, when predictions were shared across seven or more models.

Although this paper did not detail the comparison of CFS and GWI from a sedentary control, these results are available in prior published material [24,25] and the comparison of the regions selected by these models to this model is available in the Appendix A of this paper (Appendix A).

Thirty regions remained in the ensemble model on the first day before exercise and thirty-three regions remained in the ensemble model on the second day after exercise. More AAL regions were involved in the differentiation of CFS from GWI after exercise than before exercise. The qualitative assessment of the activated regions (Figure 2) demonstrated this as well. Interestingly, seventeen regions were shared by the final pre- and post-exercise ensemble models. This was consistent with our prior studies, which had found ten regions shared between the pre- and post-exercise models for CFS [24] and GWI [25]. This suggests there might be some underlying activation patterns that differentiate CFS and GWI whether or not the subject has exertional exhaustion.

The accuracy in the regions selected by all models (the overlap) improved in differentiating the two conditions by almost 6%. This indicated that it is possible there are mechanisms that differ between the two conditions that cause differentiated activation between CFS and GWI. Although post-exertional malaise is experienced by both groups of subjects, it is possible the average subject of either condition experiences it for different underlying reasons that translate to differing activation in BOLD results.

It is important to note that the variable feature importance identified by the Random Forest shows only how important those features are to that model. It can be inferred that an important variable in a highly predictive model is also important for the prediction of the condition being tested. However, there could exist indicators of the respective conditions that would show strong importance in another model, or indicators that would show strong importance for a different task rather than the n-back task. More testing and further studies would help to determine this.

The symptom profiles of CFS and GWI were very similar (Table 1) and the magnitude of differences in symptom profiles between the two was very small. The final model features reflected this, as the insula and parietal regions, which are key components of the “pain matrix”, remained as features in the models on both days. However, the insula, parietal, somatosensory strip, and other regions of the “pain matrix” that differentiated GWI and CFS from a sedentary control in our previous studies were not consistently part of the models [23,24,25]. Interestingly, the orbitofrontal cortex was in models on both days and is involved in reward, effort, and fatigue [55]. This suggests the hypothesis that differences in the processing of fatigue and reward may distinguish between CFS and GWI. However, the multivariate model does not implicate any individual brain region and so cannot implicate specific activities related to fatigue, reward, pain, or affective responses. The parietal regions remained in the models on both days suggesting that their function in salience, executive control, default, dorsal, and ventral attention networks may differentiate between CFS and GWI. The AAL atlas is not particularly helpful for defining nodes from these networks because the AAL regions are large and do not exactly correspond to the small nodes from these networks that can be defined from BOLD and functional connectivity studies. For example, the AAL parietal lobe regions encompass several of the parietal nodes from these networks. The dorsolateral prefrontal cortex, involved in the frontal parietal executive control network, does not correspond to any one or group of AAL regions. Therefore, it can be easy to overinterpret speculations of functional connectivity mapped to AAL regions. Other studies have demonstrated differences in the parietal regions using BOLD data [23] and should be further investigated to define smaller selective ROIs. While it is unlikely that the minimal subjective differences in symptom profiles will identify characteristics that catalogue GWI versus CFS, the objective patterns of differences found using the machine learning approach offer a more statistically defensible approach to define the physiological underpinnings of disease mechanisms.

This study was limited both due to its small sample size and the decision to use an ensemble modeling technique. The small sample size and use of techniques such as a Random Forest bias the results of the model towards potential overfitting. This could be rectified with more data and further studies. The use of an ensemble model can lead to results that do not translate well to future models if generated from models with high variance, if generated from models that overfit the data, or if generated from datasets that are biased in other ways. The use of bagging, when applied to regions of high variance, can lead to degradation in performance for future models. Evaluating the average performance across a series of models and of the accuracy on the region of overlap helped to mitigate some of these problems and identify AAL regions that performed best across multiple inputs. An ensemble model also requires more computational power and downstream analysis to combine the results than evaluating any one candidate model. Future studies could take advantage of a dynamic functional connectivity approach using a data-driven topological filtering approach as well.

This combined modeling technique and study was also limited because the increase in accuracy led to a decrease in the sensitivity of the model. There are many potential explanations for this drop. It is possible that the small sample size is the cause of this decrease in sensitivity, that the model is only able to select some at a highly accurate rate, or that CFS itself is a too widely variable disorder to be singularly distinguished from GWI. The Committee on the Diagnostic Criteria for Myalgic Encephalomyelitis/Chronic Fatigue Syndrome had previously found that the criteria for the diagnosis of CFS is very broad, and it is possible for two subjects to both receive the diagnosis despite having very little symptom overlap [5]. There may exist subsets of different subset phenotypes within CFS that cause subjects to have different patterns of activity on an fMRI image.

It is also important to note that no individual model outperformed the results of the combined selections found in all regions of the overlap between models. The use of multiple models both helped to increase accuracy and decrease the potential for overfitting.

## 5. Conclusions

An ensemble model used on a set of AAL regions produced from an fMRI of Chronic Fatigue Syndrome (CFS) and Gulf War Illness (GWI) was able to differentiate the two disorders before and after exercise. The results of this study indicate that use of multivariate modeling techniques could be clinically useful to differentiate the two conditions. Future studies could benefit from a larger dataset and more testing.

## Figures and Tables

**Figure 1 brainsci-10-00456-f001:**
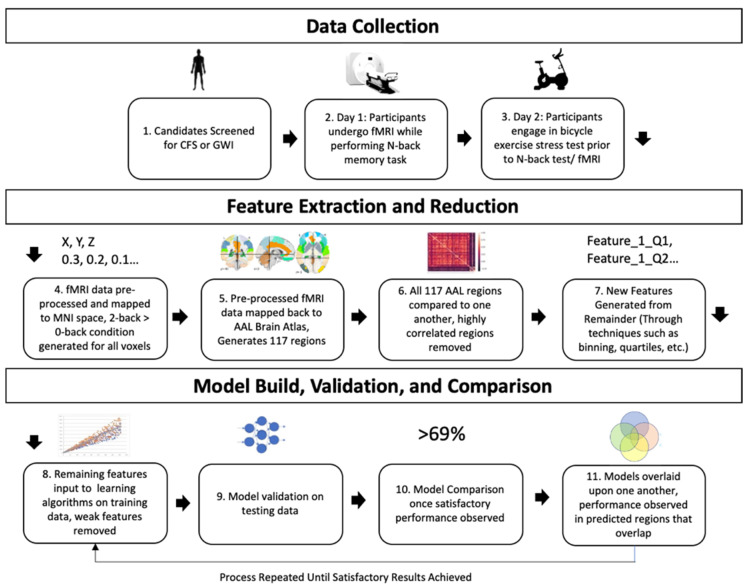
Overview of data collection methodology and model build. It is important to note that although features were generated across all data, all model builds, feature reductions, and eliminations were conducted only on the training set. Models were validated on a separate “hold-out” testing set. Models were iteratively rebuilt until the best models remained. Model performance was then combined by overlaying predictions on the testing set, these regions of overlap were then compared to determine the best overall combination of models and to create a strong predictive capacity.

**Figure 2 brainsci-10-00456-f002:**
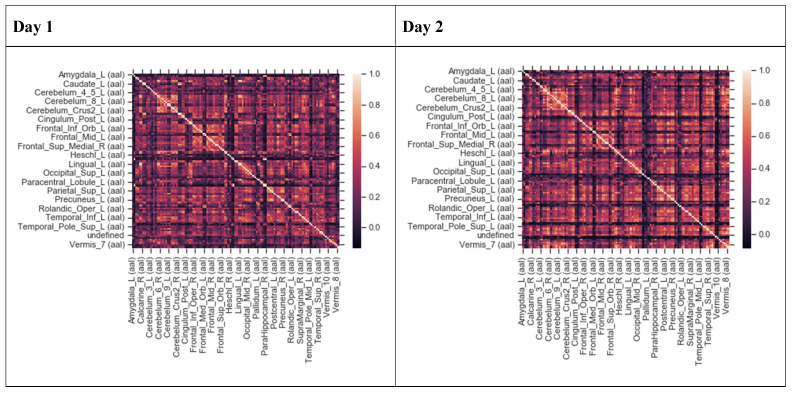
Heat maps depicting Pearson’s correlation coefficients (R) for all Automatic Anatomical Labeling (AAL) regions in Chronic Fatigue Syndrome (CFS) and Gulf War Illness (GWI) datasets for Day 1 and Day 2. The diagonal white line indicates R = 1.

**Figure 3 brainsci-10-00456-f003:**
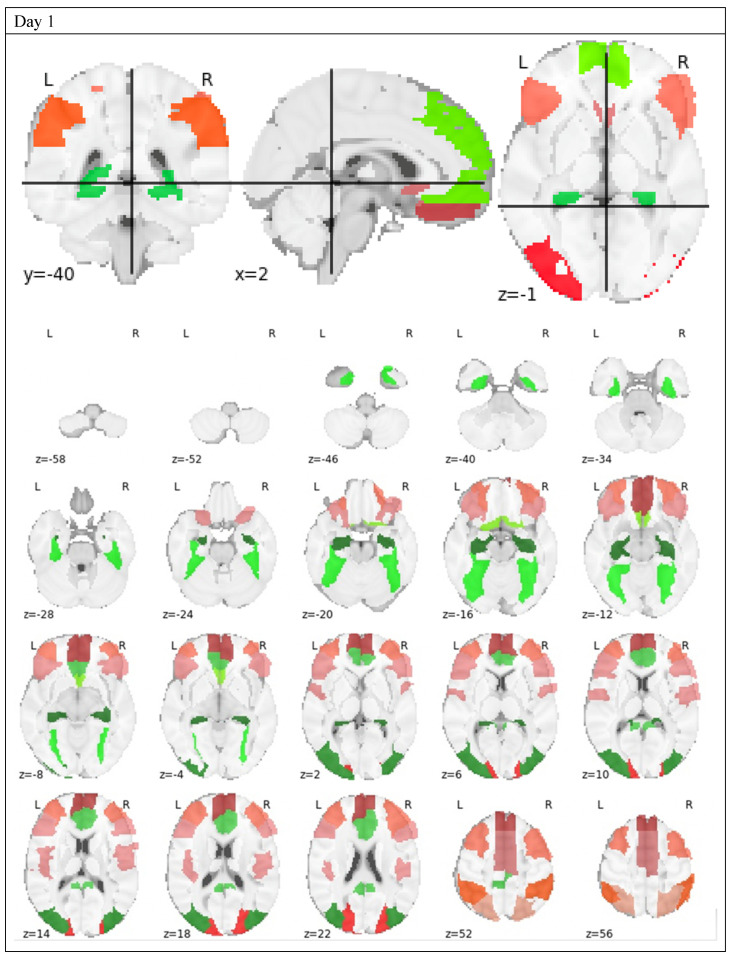
AAL regions selected across Bucket of Models and plotted on the brain for qualitative evaluation. Each of these AAL regions was selected by one or both of the respective Day 1 or Day 2 models based on the training set and validated to achieve the importance metric on the testing set.

**Figure 4 brainsci-10-00456-f004:**
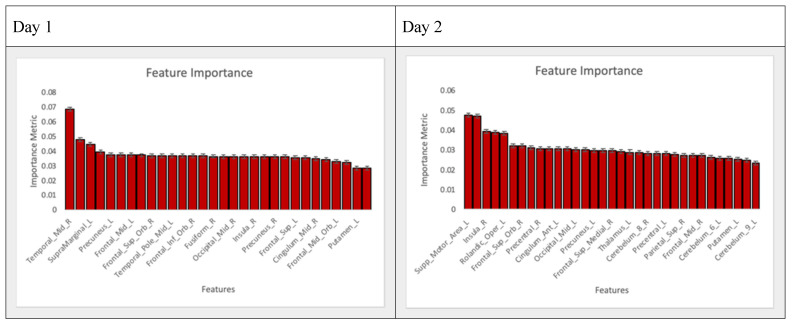
Histogram of feature importances based on Random Forest for all features. High error bars were coerced from a small dataset and repeated testing on various models. Features are plotted in ranked order.

**Figure 5 brainsci-10-00456-f005:**
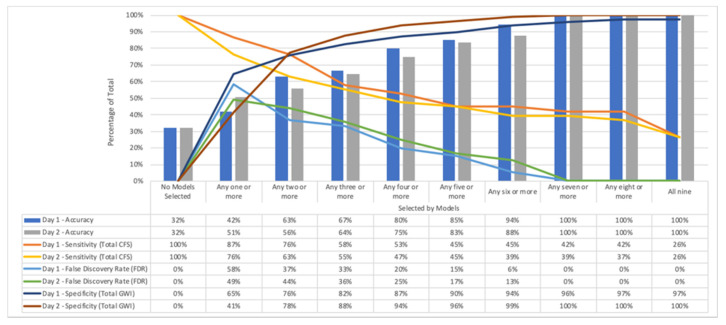
Total model accuracy of selections plotted against total number of models that selected that particular segment. Accuracy, sensitivity, and False Discovery Rate (FDR) are reported. Segments are considered as the group of selections (one or more, two or more, etc.) and accuracy was defined according to the number of correct predictions of CFS vs. GWI. Sensitivity was calculated for the total CFS selected based on the total population of CFS for each segment (the remainder are GWI). Sensitivity decreased as accuracy increased, which was one limitation of the combined modeling technique. Specificity is reported, however, special concern should be given to the artificially high specificity of the higher models and the problematic nature of determining if a negative was a true negative or false positive not selected by any segment once more models were combined. There was a general positive increase in accuracy based on total number of models that selected the population. Including all selections from any one model increased the total number of false positives in the selected segment and decreased the accuracy.

**Figure 6 brainsci-10-00456-f006:**
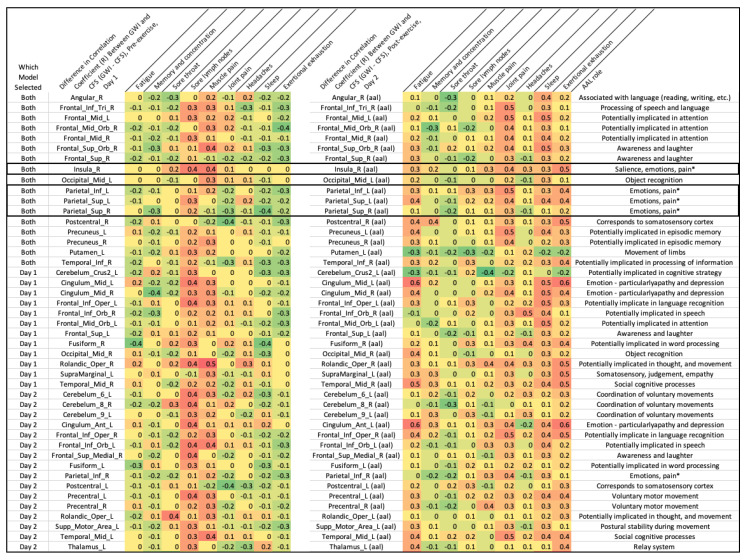
Differences between GWI and CFS in Pearson’s correlation coefficients (symptom severity score vs. AAL region) were compared on both days. These differences were also compared to prior known functional associations to AAL regions. Most notably, regions implicated in the “pain cortex” (parietal cortex and insula) had larger differences in correlation coefficients between GWI and CFS [23,24,25]. Many of the remaining regions selected by the model play a role in attention and memory even if the individual difference in correlation was not strong. This is likely explained by the use of these regions for the n-back working memory task.

**Table 1 brainsci-10-00456-t001:** Demographics (mean ± SD).

Group	GWI	CFS
N	80	38
Age	46.9 ± 7.8	47.74 ± 16.46
BMI	29.6 ± 5.6	26.20 ± 4.52
Male	59 (73.8%)	10 (26.3%) ^†^
White	64 (80.0%)	34 (89.4%) ^†^
CFS Symptom Severity Scores *^,†^		
Fatigue	3.5 ± 0.7	3.4 ± 0.8 **
Memory and concentration	3.1 ± 0.8	2.9 ± 0.9 **
Sore throat	1.4 ± 1.2	1.0 ± 1.0 *
Sore lymph nodes	1.5 ± 1.3	1.0 ± 1.1 *
Muscle pain	3.1 ± 1.0	2.5 ± 1.3 **
Joint pain	3.2 ± 1.0	1.8 ± 1.4 *
Headaches	2.7 ± 1.2	2.0 ± 1.3 *
Sleep	3.5 ± 0.8	3.2 ± 0.9 **
Exertional exhaustion	3.3 ± 1.0	3.5 ± 0.8 **

* trivial, ** mild, ^†^ False Discovery Rate (FDR) < 0.05 for each item to correct for multiple comparisons.

**Table 2 brainsci-10-00456-t002:** Features selected across Bucket of Models and corresponding importance metrics as determined by Random Forest, and coefficients as determined by a Linear Support Vector Machine (SVM) kernel and Logistic Regression. Each of these AAL regions was selected by one or both of the respective Day 1 or Day 2 models based on the training set and validated to achieve the importance metric on the testing set.

		Day 1 (Pre-Submaximal Exercise)			Day 2 (Post-Submaximal Exercise)		
	Feature	Random Forest Importance	SVM (Linear Kernel)	Logistic Regression	Random Forest Importance	SVM (Linear Kernel)	Logistic Regression
Observed in Both	Angular_R	−0.04	0.03	0.08	−0.03	0.01	0.04
	Frontal_Inf_Tri_R	−0.03	−0.01	−0.02	−0.04	−0.06	−0.2
	Frontal_Mid_L	−0.04	0.01	0.04	−0.02	0.01	0.06
	Frontal_Mid_Orb_R	−0.03	0.04	0.18	−0.03	−0.02	−0.08
	Frontal_Mid_R	−0.03	−0.01	−0.02	−0.03	0	−0.02
	Frontal_Sup_Orb_R	−0.03	−0.18	−0.59	−0.02	−0.01	−0.1
	Frontal_Sup_R	−0.03	0.02	0.05	−0.03	0.04	0.16
	Insula_R	−0.03	0.03	0.13	−0.02	0.07	0.26
	Occipital_Mid_L	−0.03	−0.03	−0.07	−0.03	−0.02	−0.1
	Parietal_Inf_L	−0.04	0	−0.01	−0.03	−0.02	−0.06
	Parietal_Sup_L	−0.03	0.01	0.04	−0.03	−0.02	−0.09
	Parietal_Sup_R	−0.04	−0.01	−0.02	−0.05	−0.02	−0.06
	Postcentral_R	−0.03	0.05	0.13	−0.03	0.05	0.21
	Precuneus_L	−0.03	0.01	0.01	−0.04	−0.04	−0.13
	Precuneus_R	−0.03	−0.01	−0.05	−0.04	0.01	0.04
	Putamen_L	−0.03	0.2	0.81	−0.03	−0.02	−0.08
	Temporal_Inf_R	−0.03	0	0.03	−0.03	0.01	0.07
Day 1 Only (Before Exercise)	Cerebellum_Crus2_L	−0.04	0.01	0.02			
	Cingulum_Mid_L	−0.04	0.05	0.15			
	Cingulum_Mid_R	−0.04	−0.08	−0.28			
	Frontal_Inf_Oper_L	−0.04	−0.02	−0.08			
	Frontal_Inf_Orb_R	−0.04	0	−0.02			
	Frontal_Mid_Orb_L	−0.04	−0.13	−0.44			
	Frontal_Sup_L	−0.03	0.02	0.06			
	Fusiform_R	−0.03	0	−0.04			
	Occipital_Mid_R	−0.03	−0.08	−0.23			
	Rolandic_Oper_R	−0.03	0.01	0.14			
	SupraMarginal_L	−0.03	−0.01	−0.07			
	Temporal_Mid_R	−0.04	0.01	0			
	Temporal_Pole_Mid_L	−0.03	0.01	0.11			
Day 2 Only (After Exercise)	Cerebellum_6_L				−0.03	0.08	0.32
	Cerebellum_8_R				−0.02	0.02	0.12
	Cerebellum_9_L				−0.03	0.1	0.35
	Cingulum_Ant_L				−0.02	−0.02	−0.03
	Frontal_Inf_Oper_R				−0.03	0.02	0
	Frontal_Inf_Orb_L				−0.02	0.08	0.3
	Frontal_Sup_Medial_R				−0.03	−0.16	−0.59
	Fusiform_L				−0.03	−0.04	−0.11
	Parietal_Inf_R				−0.03	−0.01	−0.02
	Postcentral_L				−0.04	0.06	0.23
	Precentral_L				−0.03	−0.02	−0.09
	Precentral_R				−0.03	−0.03	−0.08
	Rolandic_Oper_L				−0.03	−0.06	−0.23
	Supp_Motor_Area_L				−0.05	0.03	0.12
	Temporal_Mid_L				−0.03	0.03	0.07
	Thalamus_L				−0.04	0.02	0.08

Seventeen of these regions were selected by models on both days (Angular_R, Frontal_Inf_Tri_R, Frontal_Mid_L, Frontal_Mid_Orb_R, Frontal_Mid_R, Frontal_Sup_Orb_R, Frontal_Sup_R, Insula_R, Occipital_Mid_L, Parietal_Inf_L, Parietal_Sup_L, Parietal_Sup_R, Postcentral_R, Precuneus_L, Precuneus_R, Putamen_L, and Temporal_Inf_R). This suggested these regions consistently differentiate CFS from GWI before and after exercise.

**Table 3 brainsci-10-00456-t003:** Accuracy from the highest performing overall model. Although individually selected regions may have performed higher on different regions, this model produced the best overall accuracy across all models and one of the highest accuracies (82% for Logistic Regression and Decision Tree) on the validation set overall.

Models	Day 1 (Pre-Submaximal Exercise) Accuracy	Day 2 (Post-Submaximal Exercise) Accuracy
K-Nearest Neighbors	70%	81%
Linear SVM	70%	77%
Decision Tree	82%	82%
Random Forest	77%	78%
AdaBoost	69%	81%
Naïve Bayes	74%	78%
Quadratic Discriminant Analysis (QDA)	73%	75%
Logistic Regression	82%	82%
Neural Net	76%	77%
Average	75% ± 5%	79% ± 2%

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
