# Peer review of "Machine Learning Detects Pattern of Differences in Functional Magnetic Resonance Imaging (fMRI) Data between Chronic Fatigue Syndrome (CFS) and Gulf War Illness (GWI)"

_brainsci, 2020, doi:10.3390/brainsci10070456_

Round 1

Reviewer 1 Report

This manuscript describes a human study to discriminate Gulf War Illness (GWI) from Chronic Fatigue Syndrome (CFS) by analyzing functional MRI (fMRI) images with machine learning programs to build ensemble models.  The authors have previously used this approach to distinguish each condition from sedentary controls, and now wish to differentiate subjects with the conditions without involving control groups.  Separate models were created for pre-(Day 1) and post-(Day 2) exercise periods by splitting data into training and testing sets on the basis of blood oxygenation levels across brain regions.  The authors report that ensemble models were able to distinguish the two disorders before and after exercise, demonstrating that this task could be achieved using multivariate modeling techniques.  While the authors have reported a clinically important finding, the results constitute only an incremental advancement over previously reported evidence from this group.  The following issues require attention.

  • In general, the results of this investigation are not presented in a manner that would be comprehended by a significant portion of the journal readership. The use of machine learning approaches to analyze fMRI images for discrimination of activated brain regions will not be appreciated by most readers unless the presentation can be simplified a bit without losing appropriate methodological detail.
  • There are no designations in Table 2 to indicate which features, importance metrics, or coefficients are important or notable. Perhaps the 17 regions that consistently discriminated CFS vs. GWI could be highlighted.  As it stands, Table 2 provides much non-significant information.
  • In Figure 2 the brain slice images are too small and poorly resolved, and the brain region color coding not sufficiently distinct to clearly convey useful information. Also, the numerical labels on the slice images are too small to be legible.  Furthermore, there are no apparent pre- and post-exercise or CFS vs GWI distinctions shown in Figure 2 as asserted in lines 320-324.  The authors must consider other ways that this information can be conveyed in a more reader-friendly fashion.
  • The axes of the histograms in Figure 3 have no labels. Specifically, the abscissa axis numbers overlap and have no apparent meaning.  Error bars are mentioned in the figure caption but are not shown on the graphs.

Other points:

---The sentence in lines 3-4 of the Abstract is an overstatement without a supporting reference.

---Abbreviations must be defined at the point of first use (TR/TE in line 144, KNN in line 216), or otherwise their use is highly cryptic.  What is the correct abbreviation for “field-of-view” in section 2.4?

---The new information to be reported in this paper must be more clearly defined in lines 56-58.  In its current state the text is a bit vague.

---The meaning of the descriptor “greedy” in line 198 is not apparent.

---A Figure number is needed in line 248.

---It is not apparent where the 25% and 21% false positive values came from in lines 294-295.  They do not match the values on the graph.

Author Response

The authors would like to start by thanking the reviewer for all of the time and attention to the matter and for reviewing our paper so thoroughly. Thank you also for the helpful comments that have improved the quality of the paper.

  • In general, the results of this investigation are not presented in a manner that would be comprehended by a significant portion of the journal readership. The use of machine learning approaches to analyze fMRI images for discrimination of activated brain regions will not be appreciated by most readers unless the presentation can be simplified a bit without losing appropriate methodological detail.
  • The authors appreciate the reviewer indicating that more explanation is needed to simplify this for a broader audience, and that some jargon needed to be clarified. To simplify the explanation the following updates were included with hopes of clarifying the following points of potential confusion:
  • To indicate why multiple metrics were used and how this assisted in accounting for sample imbalance this was added to the Background:
  • [Background] Model results were combined, analyzed for statistical measures such as sensitivity and the false discovery rate (FDR), and subjected to a shuffle test to ensure performance. Multiple metrics were used to further account for sample imbalance. 
  • To explain how this approach was novel and further clarify what the final analysis of multiple models attempted to accomplish, the following was added to the methods - approach section:
  • [Methods - Approach] Predictions generated this way are next typically combined through “Stacking,” where one final combining algorithm is used to generate predictions. Instead of using one final predictive “combiner” algorithm, our approach modified this procedure to use all of the algorithms in tandem and investigate the accuracy within regions of overlap. Here a region of overlap, or segment of subjects, was defined to be predictions selected by one or more models.
  • To further explain why two datasets were used and how the model build dataset differed than the validation set, the methods section was updated to the following:
  • [Methods - Feature Selection] The testing set here functions as a validation, or hold out sample, that is different and does not overlap with the training sample used to build the model. Thus the training data used to build the model was kept separate from the validation/ testing data used to report the results. The dataset was tested at several partitions for each modeling “bucket” at a respective 70:30, 60:40, and 50:50 split into training and testing. These ratios were tested and data was iteratively resampled to also account for any biases caused by sample imbalance during modeling. Ultimately it was decided to use the 50:50 split as it provided ample data to validate and produced statistically significant and predictive model results.
  • To further explain what is meant by RFE and logistic and what a greedy elimination method is, the following was added to the methods section:
  • [Methods - Feature Selection]A greedy algorithm, like RFE or a Logistic Regression, adds and removes new features until it determines the best local optimized model at that point in time.
  • To further explain what stacking is and why the models were combined in the matter they were, the following description was added to the method section:
  • [Methods - Feature Selection]This approach, similar to a “Bucket of Models” approach, allowed for both high accuracy and generalizability as models created from features used in multiple models generally outperformed any single model working alone. Next, in an approach similar to "Stacking," predictions were compared across different models to determine the accuracy within regions of overlap, or segments of subjects.
  • There are no designations in Table 2 to indicate which features, importance metrics, or coefficients are important or notable. Perhaps the 17 regions that consistently discriminated CFS vs. GWI could be highlighted.  As it stands, Table 2 provides much non-significant information.
  • The authors appreciate the reviewer indicating the need for clarification. Table 2 has been split to indicate which regions indicate CFS vs GWI before exercise only, after exercise only, or both. We agree this improves the quality of the paper and ease with which Table 2 can be interpreted. 
  • In Figure 2 the brain slice images are too small and poorly resolved, and the brain region color coding not sufficiently distinct to clearly convey useful information. Also, the numerical labels on the slice images are too small to be legible.  Furthermore, there are no apparent pre- and post-exercise or CFS vs GWI distinctions shown in Figure 2 as asserted in lines 320-324.  The authors must consider other ways that this information can be conveyed in a more reader-friendly fashion.
  • The authors thank the reviewer for indicating the difficulty in interpretation of this figure and suggestions for improvement. The figure was remade to be larger, and recolored and labeled in a manner to make it easier to interpret. The slices now are colored shades of red/orange to represent selection by both models, green to represent Day One (Pre-exercise) only, and blue to represent Day two (Post-exercise only). The models are presented sequentially with colorings for Day One and Both grouped together, and Day Two and Both grouped together. The new figure can be found in the manuscript.
  • The axes of the histograms in Figure 3 have no labels. Specifically, the abscissa axis numbers overlap and have no apparent meaning.  Error bars are mentioned in the figure caption but are not shown on the graphs.
  • The authors recognize that this could be confusing and appreciate the reviewer pointing out this lack of labels and differentiators on the chart. The charts have both been remade. New labels are included, new axis titles are included, the charts are larger in respect to the text, and the color of the bar chart was changed to make the error bars visible. The new figure can be found in the manuscript.

Other points:

---The sentence in lines 3-4 of the Abstract is an overstatement without a supporting reference.

  • The authors appreciate the reviewer calling to attention this statement in the abstract. This concept has been referenced previously in other papers. We reference this concept and the papers that state this in lines 48- 49 of the background. The following are the citations that speak to it:
  • Pichot, P. (1994). Neurasthenia, yesterday and today. Encephale20(Spec No 3), 545–549.
  • Pearce, J. M. (2006). The enigma of chronic fatigue.  Neurol. 56, 31–36. doi: 10.1159/000095138

---Abbreviations must be defined at the point of first use (TR/TE in line 144, KNN in line 216), or otherwise their use is highly cryptic.  What is the correct abbreviation for “field-of-view” in section 2.4?

  • The authors appreciate the reviewer recognizing this oversight in regards to abbreviations. Abbreviations have been changed to reflect an explanation upon first use. In addition, KNN has been changed to exclusively reflect use of “K Nearest Neighbors” instead of KNN at some points and Nearest Neighbors at others to clear up any further potential confusion. The following additions have been made to the manuscript:
  • [Abstract] Results: A K Nearest-Neighbor (70%/ 81%), Linear SVM (70%/77%), Decision Tree (82%/82%), Random Forest (77%/78%), AdaBoost (69%/81%), Naive Bayes (74%/78%), QDA (73%/ 75%), Logistic Regression model (82%/ 82%), and Neural Net (76%/ 77%) were able to differentiate CFS from GWI before and after exercise with an average of 75% accuracy in predictions across all models before exercise and 79% after exercise.
  • K Nearest Neighbor algorithm, Linear SVM, Decision Tree, Random Forest, AdaBoost, Naive Bayes Classifier, QDA, Logistic Regression, and Neural Net.
  • Repetition Time/ Echo Time (TR/TE)= 1900/2.52ms, T1= 900ms,
  • field-of-view(FoV)=250mm,
  • We passed features through a K Nearest Neighbor (KNN) algorithm, Linear Support Vector Machine (SVM), Decision Tree, Random Forest, AdaBoost, Naive Bayes Classifier, Quadratic Discriminant Analysis (QDA), Logistic Regression, and Neural Net (Multilayer Perceptron) to create an ensemble model that generated a higher predictive power across regions selected by all of the above.
  • A K Nearest Neighbors algorithm (Day 1: 70%/ Day 2: 81%), Linear SVM (Day 1: 70%/ Day 2: 77%), Decision Tree (Day 1: 82%/ Day 2: 82%), Random Forest (Day 1: 77%/ Day 2: 78%), AdaBoost (Day 1: 69%/ Day 2: 81%), Naive Bayes (Day 1: 74%/ Day 2: 78%), QDA (Day 1: 73%/ Day 2: 75%), Logistic Regression model (Day 1: 82%/ Day 2: 82%), and Neural Net (Day 1: 76%/ Day 2: 77%) were able to differentiate CFS from GWI before and after exercise, respectively with an average of 75% accuracy in predictions across all models before exercise and 79% after exercise. (Table 3, Table S2)
  • [Table 3] K Nearest Neighbors

---The new information to be reported in this paper must be more clearly defined in lines 56-58.  In its current state the text is a bit vague.

  • The authors appreciate the reviewer indicating the need for clarity in explaining the purpose of the paper. The following update has been made to the manuscript:
  • [Background] The outcome of this model was a multivariate pattern of AAL regions activated in BOLD fMRI data that differentiate CFS from GWI. This difference indicates that despite similar symptoms, they have a pattern of different BOLD activation patterns that suggest they are separate diseases.

---The meaning of the descriptor “greedy” in line 198 is not apparent.

  • The authors recognize this terminology may not be common knowledge to the general public and will explain it further as it is a specialized term specific to machine learning. A greedy algorithm generally means it is an algorithm that sequentially adds and removes new features until it determines the most optimal solution at that point in time. Although it may not be the best global solution, the algorithm will terminate when it finds the local point of optimization. The following text has been added to the manuscript:
  • [Methods] A greedy algorithm, like RFE or a Logistic Regression, adds and removes new features until it determines the best local optimized model at that point in time.

---A Figure number is needed in line 248.

  • The authors apologize for this oversight and thank the reviewer for identifying this. A label has been - added to the text. The line at 248 now reads “Figure 1.”

---It is not apparent where the 25% and 21% false positive values came from in lines 294-295.  They do not match the values on the graph.

  • The author’s appreciate the reviewer pointing out this problematic terminology and apologize for this oversight. The 25% and 21% were calculated as the average false positive rate. The text has been updated to reflect that this simply means it could be a false positive. The text now reads the following:
  • Although the average accuracy was 75% on Day 1 and 79% on Day 2 across all models, it can be inferred that if a region was selected by only one model it might be more likely to be a false positive selection.

Reviewer 1, thank you again so much for your time and attention to the matter. On behalf of all the authors, we really appreciate your review and comments. 

Reviewer 2 Report

The manuscript by Provenzano applied machine learning techniques to accurately differentiate between patients of Gulf War Illness (GWI, n = 80) and Chronic Fatigue Syndrome (CFS, n = 30). While the manuscript demonstrated great synergy in combining neuroscience with computational techniques, I provide several comments to strengthen the manuscript:

  1. Most of the manuscript described in detail computational techniques involved to differentiate GWI and CFS. Yet, there seems a lack of novelty in applied computational methods, apart from applying routine ensemble machine learning algorithms. Can the authors highlight what is novel in their computational approach?
  2. It seems just by classifying demographic information, one may be able to accurately differentiate between GWI and CFS (which was the case for an ADHD200 competition). Have the authors tried comparing classification using demographic information alone and see what additional information brain data provide?
  3. For the neuroscience component, no explanations have been provided linking the predictive features to the underlying function of these regions and there were no discussions as to why these particular regions would differentiate between two disorders. 
  4. It seems to me that the author did not have a holdout sample that was not used to tune the models. Many early machine learning research indeed reports results based on the same sample that the models were built on. This practice is inappropriate. If there is no hold-out sample, the accuracy (especially the 100% authors obtained) is likely much inflated.
  5. No justification was provided as to why the authors chose a working memory paradigm to apply machine learning techniques, rather than say, resting-state or structural MRI. This seems to be done just out of convenience.
  6. It is slightly unclear if the activation voxels are chosen per participant or on a group basis. If they are chosen on a group basis, are the group analyses conducted based on all data (training and testing), or just on the training data?
  7. There was a lot of focus recently on using the connectome rather than activations to predict behavior. Since the authors used CONN, have the authors tried using the connectome to classify? How did the authors decide to use activations rather than connectome?
  8. The machine learning algorithms aim to differentiate GWI vs. CFS, but did the author account for the apparent sample imbalance (ns = 80/30)?
  9. The abstract states that there are 30 CSF patients but L136 and Table 1 states 38. Please keep it consistent (only report the ones that have MRI data).
  10. It is unclear why the specificity can’t be determined in Figure 4, but can be derived for different models in Table S2. As the classification is basically “A or B”, there should not be ambiguities.

Minor:

  1. Several figures are of low resolution and not legible
  2. There are double periods at the end of a few sentences
  3. Should be plus/minus sign for column 2 in Table 1
  4. Should be “two sample t-test” not “unpaired”
  5. Fonts and formats are not consistent in several places
  6. L163: Xjview not xiview
  7. L100 “easily generalizability and accessibility”

Author Response

The authors would like to begin by thanking the reviewer for their time, attention, comments, and suggestions for improvement. 

  1. Most of the manuscript described in detail computational techniques involved to differentiate GWI and CFS. Yet, there seems a lack of novelty in applied computational methods, apart from applying routine ensemble machine learning algorithms. Can the authors highlight what is novel in their computational approach?
    1. The authors appreciate the reviewer indicating there might be some confusion as to what is different about this approach and appreciate the attention called to it. The novelty of our paper lies in the application of the routine ensemble models (CFS vs GWI), the method of data analysis and pre-processing for feature selection (AAL atlas to segment and select regions of the brain), the combination of the models (multiple models in tandem), and the final calculation of accuracy (1 or more, 2 or more, 3 or more, etc.). We agree with the reviewer that it certainly would add strength to the paper to clarify why our combinative approach of the routine models in particular is novel. In particular most ensemble models seek to use one final “combiner,” while for this study we opted to use multiple models and seek to find the regions of overlap to create a better predictive capacity that outperformed any one combiner alone.
    2. The authors would also like to stress the importance of the novelty of the application itself within the differentiation of CFS from GWI. This does present a method to differentiate two disorders that don’t always have a clear path to diagnosis and present a method that might allow so. The following text has been added to further clarify the novelty in the approach itself:
    3. [Methods] To find the best optimal final Bucket of Models, we sought to report the combination of AAL regions that resulted in a threshold of 69% or greater final accuracy in predictive power of GWI vs CFS across all of the “Bucket” of models. This threshold of 69% was selected to attempt to include the results of more models and potential selected AAL regions for later filtering using the combined modeling technique.
    4. [Methods - Approach] Predictions generated this way are next typically combined through “Stacking,” where one final combining algorithm is used to generate predictions. Instead of using one final predictive “combiner” algorithm, our approach modified this procedure to use all of the algorithms in tandem and investigate the accuracy within regions of overlap. Here a region of overlap, or segment of subjects, was defined to be predictions selected by one or more models.
    5. [Methods - Model Build] This approach, similar to a “Bucket of Models” approach, allowed for both high accuracy and generalizability as models created from features used in multiple models generally outperformed any single model working alone. Next, in an approach similar to "Stacking," predictions were compared across different models to determine the accuracy within regions of overlap, or segments of subjects. A separate analysis was undertaken to determine how accuracy and generalizability changed when a segment of subjects within the validation set was selected, or predicted to be an outcome state of CFS, by one or more models. For example if a subject was selected, or predicted to be CFS, by any one model, it would be included in the one or more category. Similarly for the two or more, a subject had to be selected within the validation set by at least two models. The accuracy for each of these segments (One or more, two or more, three or more, etc.) was then compared to the total number of subjects selected in each of these segments. This was done because a highly accurate model that selected only 1% of the population would not be effective in practice. The accuracy in this analysis was determined by the total correct predictions of CFS vs GWI status.
  2. It seems just by classifying demographic information, one may be able to accurately differentiate between GWI and CFS (which was the case for an ADHD200 competition). Have the authors tried comparing classification using demographic information alone and see what additional information brain data provide?
    1. The authors thank the reviewer for this suggestion and the reference to the competition. Demographic data has been analyzed and is being compared in a different study, however was not included in this one because the symptom profiles were not substantial enough to differentiate GWI from CFS using this approach. To satisfy the reviewer’s interest - irritable bowel symptoms were worse in GWI than CFS. But no other symptom profile separated GWI from CFS, including subsets stratified by gender and Start/Stopp status (GWI exercise physiology) status. This information was not included in this study as it didn’t provide any additional insights, however the idea of stacking and comparing demographic and neurological data is certainly of merit and worthy to pursue in future studies, for this or for other disorders.
  3. For the neuroscience component, no explanations have been provided linking the predictive features to the underlying function of these regions and there were no discussions as to why these particular regions would differentiate between two disorders. 
    1. The authors appreciate the reviewer indicating the lack of functional and physiological basis or explanation for the model results. To attempt to partially clarify this, Table 2 was updated to demonstrate some importance of the features and split into sections that show which AAL regions were indicated in both models, before exercise only, and after exercise only. This analysis for the manuscript chose to adopt a strict algorithmic approach to the topic and focused on machine learning rather than the physiological underpinnings of the selections. The goal was to tease out how an algorithm, or multiple algorithms in this case, could predict physiological results such as the GWI vs CFS status. Although there is not a physiological component to the analysis, we hope to have demonstrate how one could use algorithms like this or similar to predict physiology and/or symptoms.
  4. It seems to me that the author did not have a holdout sample that was not used to tune the models. Many early machine learning research indeed reports results based on the same sample that the models were built on. This practice is inappropriate. If there is no hold-out sample, the accuracy (especially the 100% authors obtained) is likely much inflated.
    1. The authors appreciate the reviewer indicating some confusion as to whether there was a hold out sample and thank the reviewer for this point. This is a very good point as training and validating on the same sample would indeed lead to biased results. This study did use a separate training and testing (hold out) set, however in this paper we refer to the hold out sample as a testing set or validation set. All models were built on the training sample, and all results were generated from this testing/validation set. To address this and clear up any further confusion, a line was added in the following paragraph clarifying that by testing set, we mean a validation or hold out sample:
    2. Each AAL region was subjected to multiple rounds of feature selection, elimination, and then ultimate determination of importance and input into the model. The dataset was first partitioned into separate training and testing sets to build the predictive models. The testing set here functions as a validation, or hold out sample, that is different and does not overlap with the training sample used to build the model. Thus the training data used to build the model was kept separate from the validation/ testing data used to report the results. The dataset was tested at several partitions for each modeling “bucket” at a respective 70:30, 60:40, and 50:50 split into training and testing. Ultimately it was decided to use the 50:50 split as it provided ample data to validate and produced statistically significant and predictive model results.
  5. No justification was provided as to why the authors chose a working memory paradigm to apply machine learning techniques, rather than say, resting-state or structural MRI. This seems to be done just out of convenience.
    1. The authors appreciate the reviewer indicating the lack of resting state or structural MRI data and thank the reviewer for the suggestion. The authors selected the working memory paradigm for this project as it was based on preliminary results that indicated there might be a difference. Preliminary work did not suggest much difference or change in resting state. Our original hypothesis related to cognitive state and PEM so this work followed that analysis. Future studies of resting state and functional connectivity would be great and a worthy endeavor to explore.
    2. Prior papers have detailed some of the additional details surrounding this clinical trial and reasoning behind the method selection and are available here: 
      1. Rayhan RU, Stevens BW, Raksit MP, Ripple JA, Timbol CR, Adewuyi O, VanMeter JW, Baraniuk JN. Exercise challenge in Gulf War Illness reveals two subgroups with altered brain structure and function. PLoS One 2013;8:e63903. pmid:23798990
      2. Rayhan RU, Stevens BW, Timbol CR, Adewuyi O, Walitt B, VanMeter JW, et al. Increased brain white matter axial diffusivity associated with fatigue, pain and hyperalgesia in Gulf War illness. PLoS One 2013;8:e58493. pmid:23526988
      3. Baraniuk JN, El-Amin S, Corey R, Rayhan R, Timbol C. Carnosine treatment for gulf war illness: a randomized controlled trial. Glob J Health Sci 2013;5:69–81. pmid:23618477
      4. Clarke T, Jamieson J, Malone P, Rayhan R, Washington S, VanMeter J, Baraniuk J. Connectivity differences between Gulf War Illness (GWI) phenotypes during a test of attention. PLOS One. December 31, 2019. https://doi.org/10.1371/journal.pone.0226481
      5. Rayhan RU, Stevens BW, Raksit MP, et al. Exercise Challenge in Gulf War Illness Reveals Two Subgroups with Altered Brain Structure and Function. Valdes-Sosa PA, ed. PLoS ONE. 2013;8(6):e63903. doi:10.1371/journal.pone.0063903.
      6. Garner RS, Rayhan RU, Baraniuk JN. Verification of exercise-induced transient postural tachycardia phenotype in Gulf War Illness. Am J Transl Res. 2018 Oct 15;10(10):3254-3264. eCollection 2018. PubMed PMID: 30416666; PubMed Central PMCID: PMC6220213.
    3. Our manuscript references these prior studies here:
      1. Functional Magnetic Resonance Imaging (fMRI) has emerged as a promising methodology to differentiate both GWI and CFS from corresponding sedentary controls (SC) by examinations of regions that are activated or deactivated at rest or during tasks, and changes in brain activation that were caused by exertion. Our group previously established that differences persisted between GWI and a sedentary control (SC) on an fMRI scan after a mild exercise stress test.  Using a similar experimental setup, our group was also able to differentiate Chronic Fatigue Syndrome (CFS)  and GWI from a sedentary control before and after exercise using machine learning (a logistic regression algorithm). We also found that CFS was differentiable from GWI after exercise using fMRI data collected in a similar fashion. We hypothesized that using a similar experimental set up, fMRI data from both GWI and CFS and machine learning, could be used to differentiate GWI from CFS as well.
    4. And here:
      1. Subject demographics are reported here. Full subject data and performance results from the N-back working memory task between the two groups are available and published elsewhere and in online supplementary materials.
  6. It is slightly unclear if the activation voxels are chosen per participant or on a group basis. If they are chosen on a group basis, are the group analyses conducted based on all data (training and testing), or just on the training data?
    1. The authors appreciate the reviewer indicating this might be a point of confusion and thank the reviewer for this point. The activated voxels were grouped into the AAL atlas for each individual, which were then used to select features that were significant. All model build and feature selection was done on the training set, and applied to the testing set to calculate accuracy and performance. So in this way features are selected based on the individual and only on the training data. The following line was added (also referenced above in these comments) to the manuscript to help clear up this point of confusion:
    2. Each AAL region was subjected to multiple rounds of feature selection, elimination, and then ultimate determination of importance and input into the model. The dataset was first partitioned into separate training and testing sets to build the predictive models. The testing set here functions as a validation, or hold out sample, that is different and does not overlap with the training sample used to build the model. Thus the training data used to build the model was kept separate from the validation/ testing data used to report the results. The dataset was tested at several partitions for each modeling “bucket” at a respective 70:30, 60:40, and 50:50 split into training and testing. Ultimately it was decided to use the 50:50 split as it provided ample data to validate and produced statistically significant and predictive model results.
  7. There was a lot of focus recently on using the connectome rather than activations to predict behavior. Since the authors used CONN, have the authors tried using the connectome to classify? How did the authors decide to use activations rather than connectome?
    1. The authors appreciate this suggestion by the reviewer and agree that this would be an interesting matter worthy of further pursuit with more time and data. The authors did attempt to preliminarily use the connectome, however settled on activations and the grouping methodology into a well known brain atlas as there was a perceivable difference and previous studies had demonstrated this difference. The authors would like to stress that the sample size did limit the study in many aspects, and the selection of the brain atlas was certainly one of these limited in this way. If there were unlimited subjects with unlimited brain data it would certainly be easier to make inferences based on the brain atlases and connectome that group the brain into smaller segments, however for this study increasingly smaller segments resulted in regions that didn’t have enough data to be able to reliably indicate a difference. This is a very good topic for future research.
  8. The machine learning algorithms aim to differentiate GWI vs. CFS, but did the author account for the apparent sample imbalance (ns = 80/30)?
    1. The authors appreciate the reviewer indicating this imbalance and the importance of ensuring the model build is fair. Some common techniques for correcting from the sample imbalance are to use multiple evaluation metrics (to ensure for example that a population with 97% isn’t simply assuming everything falls in this category and is wrong), to resample the training set, to ensure different resampled data sets, and to resample the data with different ratios. To correct for the sample imbalance the authors reported on multiple metrics as seen in Figure 4 (Inclusion of false discovery rate in addition to accuracy), datasets were resampled during model build (As referenced in the Methods section), and multiple ratios and splits of training to testing set were explored (As referenced in the Methods section). The following text was added in this manuscript to clarify this point of confusion.
    2. [Background] Model results were combined, analyzed for statistical measures such as sensitivity and the false discovery rate (FDR), and subjected to a shuffle test to ensure performance. Multiple metrics were used to further account for sample imbalance.
    3. [Methods] These ratios were tested and data was iteratively resampled to also account for any biases caused by sample imbalance during modeling. Ultimately it was decided to use the 50:50 split as it provided ample data to validate and produced statistically significant and predictive model results.
  9. The abstract states that there are 30 CSF patients but L136 and Table 1 states 38. Please keep it consistent (only report the ones that have MRI data).
    1. The authors appreciate the reviewer indicating this typo and apologize for this oversight. The abstract has been updated accordingly.
  10. It is unclear why the specificity can’t be determined in Figure 4, but can be derived for different models in Table S2. As the classification is basically “A or B”, there should not be ambiguities.
    1. The authors appreciate the reviewer indicating that specificity could indeed be calculated. It was originally removed from the chart as it is difficult to determine if a true negative is indeed a true negative or a false positive selected by one of the models in the 9 however not used for the final analysis as it was only selected by a smaller number of models. To further explain, for the 9 models where regions selected by 1 or more models, if a region was selected by one or more, but not two or more, it could be argued to be either an FP or TN when calculating this. At the reviewer’s encouragement the authors decided to blanket assume anything not selected by the appropriate segment was a TN or FP and updated the caption to reflect this. It is also mentioned that special concern should be taken when evaluating this number as it unusually high.
    2. Figure 4: Total model accuracy of selections plotted against total number of models that selected that particular segment. Accuracy, Sensitivity, and False Discovery Rate (FDR) are reported. Segments are considered the group of selections (One or more, two or more, etc.) and accuracy was defined according to the number of correct predictions of CFS vs GWI. Sensitivity was calculated for the total CFS selected based on the total population of CFS for each segment (the remainder are GWI). Specificity is reported, however special concern should be taken to the artificially high specificity of the higher models and problematic nature of determining if a negative was a true false negative not selected by any segment once more models were combined. There was a general positive increase in accuracy based on total number of models that selected the population. Including all selections from any one model increased the total number of false positives in the selected segment and decreased the accuracy.

Minor:

  1. Several figures are of low resolution and not legible
    1. The authors appreciate the reviewer making note of this problem. The authors have updated the figures to be of higher resolution and more legible. In particular Figure 2 was made to be larger and recolored, and Figure 3 was given axis, labels, and recolored to make the error bars more visible.
  2. There are double periods at the end of a few sentences
    1. The authors appreciate the reviewer indicating this problem and apologize for this oversight. The double periods have been changed to a single period accordingly.
  3. Should be plus/minus sign for column 2 in Table 1
    1. The authors appreciate the reviewer indicating this and apologize for this oversight. The plus/ minus sign has been updated from a plus sign accordingly.
  4. Should be “two sample t-test” not “unpaired”
    1. The authors appreciate the reviewer indicating this oversight and have updated the text accordingly.
  5. Fonts and formats are not consistent in several places
    1. The authors appreciate the reviewer indicating this problem and have updated the text accordingly.
  6. L163: Xjview not xiview
    1. The authors appreciate the reviewer indicating this problem and have updated the text accordingly.
  7. L100 “easily generalizability and accessibility”
    1. The authors appreciate the reviewer indicating this problem and have updated the text accordingly. (easy instead of easily)

Thank you again for your time, attention, and comments for this manuscript. These suggestions certainly have helped improve the quality of the manuscript and we really appreciate it. 

Round 2

Reviewer 2 Report

While the authors addressed some of my comments in the response, the text is mostly unchanged and my main questions remain for this manuscript, as listed below. Therefore, I cannot recommend the publication of this manuscript.

  1. The manuscript did not integrate adequately the computational components with the underlying neural mechanisms. Up till now, it has provided little context in interpreting the results in a biologically valid way. For example, the authors found that activations in several frontal and parietal regions when patients were performing a working memory task can differentiate CFS from GWI. How can we understand this result? Does this suggest a difference in their executive functions, or in working memory functions? For a neuroscience journal, at least some interpretations are expected than just presenting results.
  2. While the authors stated they had a holdout sample, they also tried several partitions for each modeling buckets "at a respective 70:30, 60:40, and 50:50 split into training and testing" and ultimately "decided on" the 50:50 split. How the authors arrived at the decision has not been described and by repeatedly using both training and testing data to make such decisions, the independence of the testing data has already been compromised.
  3. While the accuracy is as high as 100% for the "seven or more models", the sensitivity is less than 50%. The authors need to address the low sensitivity of the promoted model.

Author Response

While the authors addressed some of my comments in the response, the text is mostly unchanged and my main questions remain for this manuscript, as listed below. Therefore, I cannot recommend the publication of this manuscript.

The authors would like to thank the reviewer for returning again with feedback, your time and effort is certainly appreciated. Substantial changes have been made to the manuscript in an attempt to address your concerns below. Please do let us know if you have additional feedback, and thank you again so much for you time and consideration. 

1. The manuscript did not integrate adequately the computational components with the underlying neural mechanisms. Up till now, it has provided little context in interpreting the results in a biologically valid way. For example, the authors found that activations in several frontal and parietal regions when patients were performing a working memory task can differentiate CFS from GWI. How can we understand this result? Does this suggest a difference in their executive functions, or in working memory functions? For a neuroscience journal, at least some interpretations are expected than just presenting results.

This is a great suggestion and the authors would like to thank the reviewer for this. The following has been added to the discussion which puts the symptom profiles, features selected, and physiological factors in context. One of the most interesting points of the results of the model is that although some aspects of the pain matrix are present in the features, the overall predictive model seems to pull from regions not linked to the GWI or CFS symptoms (reward/ fatigue regions, and salience/ executive control/ default/ attention networks). This implies that the actual causes for differentiation of CFS from GWI are due to some other physiological underpinnings that are not easily detectable through a subjective measure such as a survey. This would be very interesting to pursue in future studies to determine if CFS and GWI are in fact different and physiological based disorders. 

The following text was added to the Discussion section to address this:

The symptom profiles of CFS and GWI were very similar (Table 1) and the magnitude of differences in symptom profiles between the two was very small. The final model features reflected this, as the insula and parietal regions which are key components of the “pain matrix” remained as features in the models on both days. However, insula, parietal, somatosensory strip, and other regions of the “pain matrix” that differentiated GWI and CFS from a sedentary control in our previous studies were not consistently part of the models. [24, 25, 26, 27] Interestingly, the orbitofrontal cortex was in models on both days and is involved in reward, effort, and fatigue. [[i]] This suggests the hypothesis that differences in processing of fatigue and reward may distinguish between CFS and GWI. However, the multivariate model does not implicate any individual brain region and so cannot implicate specific activities related to fatigue, reward, pain, or affective responses. The parietal regions remained in the models on both days suggesting that their function in salience, executive control, default, dorsal and ventral attention networks may differentiate between CFS and GWI. The AAL atlas is not particularly helpful for defining nodes from these networks because the AAL regions are large and do not exactly correspond to the small nodes from these networks that can be defined from BOLD and functional connectivity studies. For example, the AAL parietal lobe regions encompass several of the parietal nodes from these networks. The dorsolateral prefrontal cortex involved in the frontal parietal executive control network does not correspond to any one single or group of AAL regions. Therefore, it can be easy to overinterpret speculations of functional connectivity mapped to AAL regions. Other studies have demonstrated differences in the Parietal regions using BOLD data [24] and should be further investigated to define smaller selective ROIs. While it is unlikely that the minimal subjective differences in symptom profiles will identify characteristics that catalogue GWI versus CFS, the objective patterns of differences found using the machine learning approach offer a more statistically defensible approach to define the physiological underpinnings of disease mechanisms.

[i] Padoa-Schioppa C, Conen KE. Orbitofrontal Cortex: A Neural Circuit for Economic Decisions. Neuron. 2017;96(4):736-754. doi:10.1016/j.neuron.2017.09.031

2. While the authors stated they had a holdout sample, they also tried several partitions for each modeling buckets "at a respective 70:30, 60:40, and 50:50 split into training and testing" and ultimately "decided on" the 50:50 split. How the authors arrived at the decision has not been described and by repeatedly using both training and testing data to make such decisions, the independence of the testing data has already been compromised.

The authors appreciate the reviewer bringing to attention this potentially confusing point and apologize for any confusion this may have caused. The multiple partitions were used to evaluate what the training: testing split should be. The ultimate deciding factor for using a 50:50 split was simply because the number of samples in the final testing set of a stratified sample at the 50:50 split allowed for more CFS subjects in the final model build. We did this test as one way to control for potential sample imbalance as the number of samples was very low. The small sample size is certainly a limitation of the study and future studies with more data would strengthen these results. The text has been updated to reflect this. If the reviewer feels this portion might confuse a reader, this could be updated to “The data was partitioned into a 50:50 training:testing set split” without mention of how we chose this split, however the authors felt it was interesting to list the methods used to account for sample imbalance to allow for greater statistical rigor of the paper.

[Methods - Feature Selection and Predictive Model Build]: Prior to model build to account for potential sample imbalance, the dataset was evaluated at several partitions at respective 70:30, 60:40, and 50:50 splits into training and testing data. This was done because of the small sample size and greater number of GWI subjects compared to CFS subjects. Sample imbalance in extreme cases can prevent the model from learning and create highly inaccurate estimates that may predict all outcomes are one class.[32] The number of samples that remained at each split was examined by looking at the total subjects in each split and by a preliminary set of test models. It was decided to use the 50:50 split as this provided ample data to validate, allowed for more CFS and GWI individuals to be present in both the training and testing set, and produced statistically significant results. These multiple “test” training and testing splits were not used in the final model build as this was a preliminary step, meaning that no data from the testing set was used to train the final set of models. The final dataset used for this manuscript was a 50:50 training to testing set split, where the training sample was used to build the model on 50% of the data, and the testing set was used to validate the model on the remaining 50%.

3. While the accuracy is as high as 100% for the "seven or more models", the sensitivity is less than 50%. The authors need to address the low sensitivity of the promoted model.

The authors appreciate the reviewer noting this potential limitation of the model and will include this in the discussion. One of the large problems with choosing the best model to propose for this paper was that models with a high accuracy and specificity often had a low sensitivity, and vice-versa. With CFS this is especially difficult as the criteria for diagnosis is very broad. The Committee on the Diagnostic Criteria for Myalgic Encephalomyelitis/Chronic Fatigue Syndrome had previously found that it’s possible for two subjects to both receive the diagnosis despite having very little symptom overlap. Other studies have shown that different CFS questionnaires can have sometimes as little as 30% overlap in symptom assessment. It’s possible that the small sample size causes this decrease in sensitivity, that the criteria for diagnosis of CFS and symptoms experienced themselves amongst the population cause this decrease in sensitivity, or that the model is simply only able to select some at a highly accurate rate. With more data it would certainly be interesting to see if there is a way to create and link subsets of CFS phenotypes to fMRI individuals (such as the START/ STOPP/ POTS phenotypes of GWI that can be linked to fMRI activations).The text has been updated to include this discussion.

[Figure 5 caption update] Sensitivity decreased as accuracy increased, which was one limitation of the combined modeling technique.

[Discussion]: This combined modeling technique and study was also limited because the increase in accuracy lead to a decrease in sensitivity of the model. There are many potential explanations for this drop. It’s possible that the small sample size is the cause of this decrease in sensitivity, that the model is only able to select some at a highly accurate rate, or that CFS itself is too widely variable of a disorder to be singularly categorized from GWI. The Committee on the Diagnostic Criteria for Myalgic Encephalomyelitis/Chronic Fatigue Syndrome had previously found that the criteria for diagnosis of CFS is very broad, and it’s possible for two subjects to both receive the diagnosis despite having very little symptom overlap. [5] There may exist subsets of different subset phenotypes within CFS that cause subjects to have different patterns of activity on an fMRI image.